# Differential Effect of the Physical Embodiment on the Prefrontal Cortex Activity as Quantified by Its Entropy

**DOI:** 10.3390/e21090875

**Published:** 2019-09-08

**Authors:** Soheil Keshmiri, Hidenobu Sumioka, Ryuji Yamazaki, Hiroshi Ishiguro

**Affiliations:** 1Hiroshi Ishiguro Laboratories (HIL), Advanced Telecommunications Research Institute International (ATR), Kyoto 619-02, Japan; 2Symbiotic Intelligent Systems Research Center, Institute for Open and Transdisciplinary Research Initiatives, Osaka University, Osaka 565-0871, Japan; 3Graduate School of Engineering Science, Osaka University, Osaka 560-8531, Japan

**Keywords:** differential entropy, embodied media, tele-communication, humanoid, prefrontal cortex

## Abstract

Computer-mediated-communication (CMC) research suggests that unembodied media can surpass in-person communication due to their utility to bypass the nonverbal components of verbal communication such as physical presence and facial expressions. However, recent results on communicative humanoids suggest the importance of the physical embodiment of conversational partners. These contradictory findings are strengthened by the fact that almost all of these results are based on the subjective assessments of the behavioural impacts of these systems. To investigate these opposing views of the potential role of the embodiment during communication, we compare the effect of a physically embodied medium that is remotely controlled by a human operator with such unembodied media as telephones and video-chat systems on the frontal brain activity of human subjects, given the pivotal role of this region in social cognition and verbal comprehension. Our results provide evidence that communicating through a physically embodied medium affects the frontal brain activity of humans whose patterns potentially resemble those of in-person communication. These findings argue for the significance of embodiment in naturalistic scenarios of social interaction, such as storytelling and verbal comprehension, and the potential application of brain information as a promising sensory gateway in the characterization of behavioural responses in human-robot interaction.

## 1. Introduction

Verbal communication is an integral element of mankind’s societal evolution whose intention is signified through such nonverbal means as facial expressions, motions, and the physical presence of communicating partners. Research suggests that a proper communication has a direct positive influence on physical and mental health [1] as well as the quality of learning [2,3]. On the other hand, Internet-based technologies [4] present easy-to-use alternatives for exchanging verbal information without nonverbal signatures. In their seminal work, Reference [5] demonstrated the tendency of individuals to treat computers and people alike in their responses which covered a wide range of social [6] and behavioural domains [7]. In fact, computer-mediated-communication (CMC) research [8] argues that the utility of unembodied media might surpass in-person communication due to its ability to bypass the nonverbal components that are inherent in face-to-face communication [9]. For instance, Joinson [10] noted that the CMC effect increases self-disclosure in individuals in contrast with in-person communication. Along the same direction, Zimmerman [11] concluded that CMC elicits emotionally richer, more relationship-oriented verbal interaction among emotionally disturbed adolescents than in-person communication. A common theme that links these findings is the elimination of the physical presence of participants (e.g., absence of visual communication) which is a significant advantage of CMC. This viewpoint finds further evidence in the growing adaptation of voice assistant agents (e.g., Siri, Cortana, Google Now, Amazon Alexa, etc.), implying individuals tend to undermine the significant influence of the physical embodiment for representing non-verbal information during verbal communication.

However, recent findings in communicative humanoids emphasize the importance of the physical embodiment of conversational partners in human-robot interaction [12]. For instance, Mann et al. [13] suggested that people are more responsive to robots than computer-based healthcare systems. Tele-operated robots (i.e., remotely controlled robotic media) seem to increase the feelings of being together in remote communication. For example, Sakamoto and colleagues [14] found that an anthropomorphic tele-operated robot improved the effect of social tele-presence more than audio- and video-conferencing. Tanaka and colleagues [15] compared the effect of a robotic medium with audio-, video-, and avatar-based media for communication and found that it enhanced the sense of a human presence at the same level as the partner’s live video stream. In addition, applications of robotic media in elderly care and intervention support their physical embodiment as a crucial factor in their effectiveness [16,17]. Broadbent [18] presented a comprehensive review of interactive robotic media.

Although these results imply the potentially positive effect of physical embodiment, almost all of them are based on the subjective assessments of the behavioural impacts of physically embodied media. This severely limits the possibility of drawing a reliable conclusion. In fact, there is a paucity of research on the neurophysiological effect of physical embodiment during verbal communication.

Previous research identified the critical role of the prefrontal cortex (PFC) in social and embodied cognition [19], given its anatomical organization that allows for the integration of information from all five senses [20]. The pivotal role of the medial (m)PFC [21] is emphasized in mediating social interaction as part of the brain’s mentalizing network [22,23], which starts from infancy [24]. In this respect, Takahashi et al. [25] showed that the mPFC modulates responses to the anthropomorphic impressions of interacting agents. Krach et al. [26] concluded that an agent’s human-likeness significantly influences mPFC activity and its corresponding mentalizing network. In fact, Chaminade et al. [27] identified the occurrence of this modulating process in response to both human-human as well as human-robot interactions. These findings argue that PFC responses are rather integrated toward physical embodiment, regardless of the nature (i.e., human, synthetic, or otherwise) of the communicating agent. However, it is crucial to note that some of the findings [25,26,27] are based on the modulation of both, the embodiment (i.e., physically situated medium) and the intelligence (human and synthetic) of their adapted media. Moreover, they mainly focused on the impact of these media in the context of interactive games. As a result, their findings fail to adequately answer more subtle questions about whose appearance is more suitable for such media, given the same level of intelligence.

Research on working memory (WM) [28] also identified PFC activation in a variety of tasks, from mental tasks with high cognitive loads [29] to such social cognition as emotional responses [30] and story comprehension [31,32]. An important implication of these results is the activation of some similar regions in PFC, regardless of the nature of the potential effect under investigation. Therefore, considering the involvement of PFC in verbal communication, as well as social and embodied cognition along with the activation of similar PFC areas in response to both auditory and embodied stimuli, it is plausible to address the utility of these neuroscientific findings to investigate other potential influences of physical embodiments on our brains during verbal communication.

We utilize these findings to study the neurophysiological effect of physical embodiment through analysis of the near-infrared spectroscopy (NIRS) [33] time series of the frontal brain activity of human subjects. We chose NIRS due to its non-invasive operational setup, portability and relative immunity to body movement, compared with other imaging techniques [34]. These advantages provide a tremendous opportunity for the adaptation of NIRS devices in naturalistic experiments such as gaming and human-robot interaction [35,36]. Although NIRS has a weaker signal-to-noise ratio (SNR) than functional magentic resonance imaging (fMRI) [37], research found that the hemodynamic responses captured as blood oxy/deoxy-genation hemoglobin highly correlate with the Blood Oxygen Level Dependent (BOLD) in fMRI [38]. This correlation is unaffected by the nature of cognitive tasks (ibid).

The human brain, like any other healthy physiological system, is an inherently complex system whose dynamics strongly correlate with the productivity of its cognitive functions such as attention and language [39]. Interestingly, an increase in complexity reflects the information content of the dynamics of physiological systems which, in turn, have a direct correspondence to the variational information in their activities [40]. Given these observations, in conjunction with PFC’s pivotal role in processing auditory and embodiment stimuli, we investigate the effect of physical embodiment on inducing potential changes in the variational information of frontal brain activity; that is, its information content or dynamics.

**Hypothesis** **1.**
*In comparison with unembodied media, a physically embodied medium induces an increase in the information content of the frontal brain activity of human subjects in response to communicated contents.*


We tested the validity of this hypothesis in a storytelling experiment in which senior citizens and adults in their 20s and 30s listened to three-minute stories from Greek mythology about constellations using three different forms of media: an audio speaker, a video-chat system, and a teleoperated robot. face-to-face was our control setting in which we communicated these stories with the participants in-person. These settings enabled us to primarily investigate the effect of physical embodiment while maintaining the same level of conveyed intelligence (i.e., human intelligence). The inclusion of video-chat also allowed us to compare any potential difference between the effect of physical embodiment as opposed to the presence [41]. It is also crucial to note that the aim of the present study was not to analyze the effect of embodiment on the human-medium interaction [42,43,44] and instead focused on verifying whether the embodiment of a medium bore a significant effect on the brain activation in response to verbally communicated content. To this end, we prevented any physical contact between the participants and the media in this study (see Section 5.2, for details).

We used near-infrared spectroscopy (NIRS) to record the frontal brain activity of the participants while they listened to the stories. They answered questionnaires about the difficulty of the narrated stories, their interest in the topics of these stories, and the degree of fatigue caused by listening to them, as well as their perceived feelings of the storyteller’s presence after each of the three types of media at the end of each storytelling session. We used self-assessed responses for three reasons. First, we used the responses to the feelings of the human presence to verify whether the potential influence of the physical embodiment is also reflected in a more immediate cognitive state of the participants. Second, we utilized the other self-assessed responses to justify whether such potential changes are associated with embodiment or whether other physiological factors are also involved, for example, fatigue, lack/loss of interest, and so forth. Last, these self-assessed responses allowed us to compare the viability of the information content of the frontal brain activity of the human subjects as a unique biomarker for the identification of the effect of physical embodiment with such behavioural signals as eye movement.

Matarić [45] adequately notes that today’s robotics impact is not merely on assisting humans perform their physical tasks but facilitating their social interaction in such a broad range of areas as child education [3,12] and elderly care [46,47]. In this regard, Yang et al. [48] consider the social interaction as one of the grand challenges of the robotics for twenty first century with modeling social dynamics, learning social and moral norms, and building a robotic theory of mind (ToM) [49] as three most significant challenges among all. It is apparent that interactive media that can comprehend their human companions’ intentions to respond effectively is the minimum requirement if such interactions and relationships are to last long. We believe that the use of neurophysiological responses such as PFC activity offers a significant complementary information for modeling the human behaviour to take the human-robot interaction research a step closer to achieving some of the missing components of truly interactive intelligent media. For instance, patterns of brain activation can form highly informative neurophysiological feedback that is otherwise hidden, even from the individuals themselves. This presents a substantial contribution to such paradigms as interactive learning in multimodal modeling of the human behaviour [50]. Furthermore, inclusion of the brain can form a stepping stone for rich and usable models of human mental state [48] and eventually a new venue to complement the behavioural- and observational-based study and analysis of a robotic ToM [49] through critical investigation of its implications in human’s neurological responses while interacting with their synthetic companions. Our results contribute to the field of human-robot interaction via providing further insights on the effect of the physical embodiment on human’s brain activation pattern, thereby taking our understandings a step closer to achieving rich and usable models of human mental state [48] and eventually a foundation for study and analysis of a robotic theory of mind [49].

## 2. Results

### 2.1. Variational Information of Frontal Brain Activity

#### 2.1.1. Left-Hemispheric PFC

Considering the effect of media and age factors, a two-way mixed-design ANOVA indicates a significant difference with respect to media (*p* < 0.001, F(1.75, 50.87) = 42.03, $\eta_{G}^{2}$ = 0.42). However, this difference is non-significant in case of age (*p* = 0.97, F(1, 29) = 0.002, $\eta_{G}^{2}$ = 0.0) and the interaction between these factors (*p* = 0.70, F(1.75, 50.87) = 1.01, $\eta_{G}^{2}$ = 0.02). Post hoc comparison with modified sequentially rejective Bonferroni correction implies significant differences between speaker and Telenoid (*p* < 0.001, t(29) = 7.61, M${}_{S}$ = 8.78, SD${}_{S}$ = 0.17, M${}_{T}$ = 9.01, SD${}_{T}$ = 0.08), speaker and face-to-face (*p* < 0.001, t(29) = 12.71, M${}_{F}$ = 9.15, SD${}_{F}$ = 0.18), video-chat and Telenoid (*p* < 0.001, t(29) = 8.08, M${}_{V}$ = 8.76, SD${}_{V}$ = 0.17), as well as video-chat and face-to-face (*p* < 0.001, t(29) = 12.28). However, it also suggests non-significant differences between speaker and video-chat (*p* = 0.26, t(29) = 1.15) as well as Telenoid and face-to-face (*p* = 0.16, t(29) = 1.42) (Figure 1a).

#### 2.1.2. Right-Hemispheric PFC

Considering the effect of media and age factors, a two-way mixed-design ANOVA indicates a significant difference with respect to media (*p* < 0.001, F(3, 29) = 9.05, $\eta_{G}^{2}$ = 0.35). However, this difference is non-significant in case of age (*p* = 0.14, F(1, 29) = 2.32, $\eta_{G}^{2}$ = 0.02) and the interaction between these factors (*p* = 0.12, F(3, 87) = 1.99, $\eta_{G}^{2}$ = 0.002). Post hoc comparison with modified sequentially rejective Bonferroni correction implies significant differences between speaker and Telenoid (*p* < 0.01, t(29) = 2.60, M${}_{S}$ = 6.32, SD${}_{S}$ = 2.05, M${}_{T}$ = 7.78, SD${}_{T}$ = 1.90), speaker and face-to-face (*p* < 0.001, t(29) = 5.02, M${}_{F}$ = 8.33, SD${}_{F}$ = 1.61), video-chat and Telenoid (*p* < 0.01, t(29) = 2.66, M${}_{V}$ = 6.28, SD${}_{V}$ = 1.20), as well as video-chat and face-to-face (*p* < 0.001, t(29) = 4.99). However, it also suggests non-significant differences between speaker and video-chat (*p* = 0.96, t(29) = 0.06) as well as Telenoid and face-to-face (*p* = 0.16, t(29) = 1.44) (Figure 1b).

### 2.2. Self-Assessed Responses to Feeling of Human Presence

A two-way mixed-design ANOVA with media and age factors implies a significant effect of media (*p* < 0.001, F(3, 87) = 43.03, $\eta_{G}^{2}$ = 0.41), with non-significant effects of age (p = 0.42, F(1, 29) = 0.68, $\eta_{G}^{2}$ = 0.01), and the interaction between these factors (*p* = 0.18, F(3, 87) = 1.65, $\eta_{G}^{2}$ = 0.03). Post hoc comparison with modified sequentially rejective Bonferroni correction implies significant differences between speaker and Telenoid (*p* < 0.001, t(29) = 6.32, M${}_{S}$ = 1.75, SD${}_{S}$ = 0.07, M${}_{T}$ = 4.77, SD${}_{T}$ = 2.06), speaker and face-to-face (*p* < 0.001, t(29) = 11.14, M${}_{F}$ = 7.17, SD${}_{F}$ = 1.68), speaker and video-chat (*p* < 0.001, t(29) = 4.74), video-chat and face-to-face (*p* < 0.001, t(29) = 6.60), and face-to-face and Telenoid (*p* < 0.001, t(29) = 5.91). However, it suggests a non-significant difference between Telenoid and video-chat (*p* = 0.60, t(29) = 0.49) (Figure 2a).

#### 2.2.1. Left-Hemispheric PFC

We found no correlation between the information content of the frontal brain activity (i.e., DE) of the participants and the EAR of their gazing behaviour in the speaker (r = 0.19, *p* = 0.32, M${}_{EAR}$ = 0.42, SD${}_{EAR}$ = 0.16), video-chat (r = 0.34, *p* = 0.06, M${}_{EAR}$ = 0.31, SD${}_{EAR}$ = 0.15), Telenoid (r = 0.17, *p* = 0.40, M${}_{EAR}$ = 0.33, SD${}_{EAR}$ = 0.16), and face-to-face (r = 0.04, *p* = 0.82, M${}_{EAR}$ = 0.46, SD${}_{EAR}$ = 0.20) (Figure 3).

Considering the responses of the participants about the difficulty of the narrated stories, their interest in them, and the fatigue caused by listening to them, a two-way mixed-design ANOVA with media and age factors is non-significant with respect to media, age, and interaction among these factors (all three self-assessments). Furthermore, the Pearson correlation results among these responses and the variational information of the frontal brain activity of the participants are non-significant (SM 2 through SM 6 for analysis details).

#### 2.2.2. Feeling of Human Presence and Left-Hemispheric Information Content

We observed a significant moderate correlation between the variational information of the frontal brain activity of the participants with their self-assessed responses to the feelings of the presence of the remote storyteller in face-to-face (r = 0.40, *p* < 0.03) and Telenoid (r = 0.40, *p* < 0.03). Interestingly, we found a significant weak anti-correlation for the speaker medium (r = −0.38, *p* < 0.05). On the other hand, it is non-significant in the case of video-chat (r = 0.04, *p* = 0.85) (Figure 2).

#### 2.2.3. Feeling of Human Presence and Right-Hemispheric Information Content

We observed a significant moderate correlation between the variational information of the right-hemispheric frontal brain activity of the participants with their self-assessed responses to the feelings of the presence of the remote storyteller in face-to-face (r = 0.38, *p* < 0.05) and Telenoid (r = 0.37, *p* < 0.05) (Figure 4). Interestingly, we found a significant weak anti-correlation for the speaker medium (r = −0.37, *p* < 0.05). On the other hand, it is non-significant in the case of video-chat (r = 0.06, *p* = 0.74).

### 2.3. Gaze Data

We found no correlation between the self-assessed responses of the participants to their feelings of the presence and EAR in speaker (r = 0.26, *p* = 0.16), video-chat (r = 0.19, *p* = 0.46), Telenoid (r = 0.16, *p* = 0.40), as well as face-to-face (r = −0.19, *p* = 0.30) (Figure 5).

#### Right-Hemispheric PFC

We found no correlation between the information content of the frontal brain activity (i.e., DE) of the participants and the EAR in speaker (r = 0.07, *p* = 0.72, M${}_{EAR}$ = 0.42, SD${}_{EAR}$ = 0.16), video-chat (r = 0.18, *p* = 0.34, M${}_{EAR}$ = 0.31, SD${}_{EAR}$ = 0.15), Telenoid (r = 0.04, *p* = 0.82, M${}_{EAR}$ = 0.33, SD${}_{EAR}$ = 0.16), and face-to-face (r = −0.25, *p* = 0.18, M${}_{EAR}$ = 0.46, SD${}_{EAR}$ = 0.20) (Figure 6).

## 3. Discussion

We studied the effect of physical embodiment on the information content of the frontal brain activity of human subjects in a storytelling experiment through comparative analyses of the effect of a physically embodied medium between video-chat and speaker. The face-to-face setting was our control setting. Our motivation for studying the frontal brain activity of the human subjects was due to neuroscientific findings on the association of this region with verbal communication that requires social cognition [19] as well as story comprehension [31,32]. Below, we discuss the implications of different aspects of our results in support of our hypothesis.

### 3.1. PFC Activation and Physical Embodiment

We observed a significant increase in the variational information of the frontal brain activity by physical embodiment. We also observed that this increase was non-significant in comparison with the in-person setting. On the other hand, the physically embodied medium induced an increase in the variational information of the frontal brain activity of the participants that differed significantly from video-chat and speaker. Additionally, the results of the ANOVA analysis implied a non-significant difference in the magnitude of this increase for both hemispheres. These results implied that face-to-face and the physically embodied medium induced a similar activation effect on the PFC of the participants.

Although the cognitive load on the working memory has a substantial bearing on PFC activation [29], it is unlikely that such an effect had a significant impact on the increased information content of the frontal brain activity of the participants. This is due to the observation that our results indicated no correlations between the variational information of the frontal activity of the participants and their responses to such feelings as difficulty of the stories and fatigue caused by listening to them. On the other hand, findings from social cognition express PFC’s pivotal role in processing the social as well as the anthropomorphic characteristics of the embodiment [19,23]. For instance, these results imply its involvement in modulating responses to the anthropomorphic impressions of interacting agents [25,26], regardless of their nature (i.e., human-human or human-robot interaction) [27]. An important observation about these findings is the modulation of both the embodiment (i.e., appearance of agents) and the intelligence (human and synthetic intelligence) of the partners during communication/interaction. This complicates the possibility of drawing a conclusion about the extent of the embodiment’s impact on the frontal activity of human subjects. Our results complemented these findings by maintaining the same level of conveyed intelligence (i.e., identical stories by the same storyteller) to solely analyze the media’s physical aspects. In such a setting, it is plausible to expect PFC to respond identically, if the media’s influence were negligible. Contrary to this expectation, our results suggest the potential utility of physical embodiment for enhancing PFC responses to physical embodiment.

Our results on the bilateral increase in the information content of the frontal brain activity in participants that are in line with recent findings on the effect of natural speech (at least within the context of storytelling) [32,51,52] point at the contribution of physical embodiment to processes related to social cognition in PFC that respond to physical embodiment and stories [23]. As for PFC roles, Forbes et al. [19] recognized three primary categories within the context of social cognition: social perceptual (processing and identification of the features of others), social categorization (effect of normative and stereotypes on perception and information processing of an individual), and attributional (understanding/interpreting the actions and the behaviours of others) processes. In this respect, the observed increase in PFC’s information content can be plausibly attributed to the perception of such social features as other’s face [19]. Although this might suggest that the human-like facial characteristic of a medium is sufficient for activation of this process, this interpretation is refuted by the lower PFC activation in the video-chat setting even though the participants were looking at the storyteller on the display. This identified the central role of physical embodiment (as opposed to mere presence) in activation of the PFC’s social perceptual process within the context of verbal communication.

In our results, the substantial effect of the situational cues of a medium on the PFC’s social categorization process is reflected in non-significant difference of the induced increase in the information content of the frontal brain activity of the participants for the physically embodied medium and in-person communication. More specifically, it implied the ability of the physical embodiment to persuade the participants to interpret the actual presence of the storyteller and her presence through the physically embodied medium under the same social categorical norm/stereotype. This observation finds further evidence in the significant correlation between the increased information content of the frontal brain activity of the participants and their feelings of the human presence in the case of the face-to-face and the physically embodied medium. On the other hand, the lower PFC’s variational information in the speaker and video-chat settings might indicate their categorization as audio-player and TV by this process.

It is apparent that the perception of the stimuli under different social categorial norms has an immediate impact on their attributed trait characteristics by the human subjects [53,54]. Forbes et al. [19] argued that the involvement of attributional processes in modulating information is pertinent to the mentalization process. Considering the controlled level of intelligence in our study, it is plausible to expect that this process contribute identically to our results. On the other hand, a previous fMRI study [25] identified the change in an individual’s impression of the embodied media with differing appearances which, in turn, indicated the different degree of the contribution of the social perceptual and the social categorization processes to PFC’s attributional process. Therefore, it is plausible to consider the observed increase in the PFC’s information content in response to the physically embodied medium and in-person as an indication of the establishment of a tentative mental association between the physically embodied medium (i.e., the other) and the participants. This, in turn, complements the results of such association processes with respect to the autonomy of the media [25,26,27].

### 3.2. PFC Activation and Feeling of Human Presence

Further evidence for the potential effect of physical embodiment on the neurophysiological responses of human subjects came from the correlational analysis of the variational information of the frontal brain activity of the participants and their self-assessed responses to their feelings of the presence of the remote storyteller. Although these responses were positively correlated for the in-person and the physically embodied medium, they implied an anti-correlation with respect to the speaker. Interestingly, the self-assessed responses of the participants showed a similar level of correlation in the physically embodied medium and in-person settings. Moreover, the negative correlation in the case of the speaker medium is an interesting observation, considering the total absence of a physical presence in this setting. Similarly, the non-significant correlation of the video-chat emphasized the crucial role of physical embodiment in producing such an effect in brain neurological responses, stressing the physical (e.g., actual presence in face-to-face and physically embodied medium for humanoids) over the visual [55] property of a medium as the primary source for the manifestation of such neurophysiological responses.

This observation accumulated further support in the non-significant responses of our participants to questionnaires regarding their emotional states with respect to the task (i.e., listening to stories), such as the difficulty of the story topic, their interest in them and the fatigue induced by listening to them (SM 2 through SM 6 for details). These results imply that the brain activity of the participants was not affected by their impressions of the conversation and/or the communicated stories.

### 3.3. PFC Activation and Gazing Patterns

Our results indicated that the gazing behaviour of the participants had no substantial contribution in the observed changes in the variational information of their brain frontal activity. This suggestion reflects that we found no correlation between the eye-movement/aversion behaviour of the participants and the information content of their frontal brain activity. Similarly, our results also indicated that the eye-movement/aversion of the participants and their self-assessed responses to the feelings of the human presence were not correlated. The latter observation suggests that the observed increase in the information content of the frontal brain activity of the participants was uniquely associated with the effect of physical embodiment and that this effect was absent in their behavioural responses.

### 3.4. PFC Activation and Aging

We observed a bilateral increase in the information content of PFC. Furthermore, our analyses indicated a non-significant effect of the age factor for both hemispheres, implying that PFC activation during physically embodied verbal communication deviated from the dedifferentiation hypothesis [56] that implies the reorganization of the PFC pattern of activation from lateralized to bilateral activation with aging. Our results suggest that PFC activation during the physically embodied verbal communication was potentially more distributed between the two hemispheres, contrary to the left-lateralized activation in mere WM tasks [57]. These results indicate that the influence of physical embodiment on the information content of the frontal brain activity of the participants was not affected by their age. They also provide a quantitative answer to the divided stances on the effect of age on the responses of individuals to physical embodiment based on their behavioural and psychological assessments [58,59]. Seider et al. [60] argued that increasing the level of cognitive activity reduces the risk of such cognitive consequences of aging as dementia. In fact, research on aging and cognition identified a substantial correlation between the increased variational information of brain activity and the dynamics of healthy cognitive functioning [39]. In this respect, our results contribute to a general theory of diminished cognitive performance with aging by focusing on the potential benefit of the adaptation of physically embodied media as a complementary cognitive training strategy to delay and/or prevent such age-related diseases as Alzheimer’s and dementia by inducing an increase in the variational information of the frontal brain activity of senior citizens [17].

## 4. Concluding Remarks

Although our results indicated the significant role of the physical embodiment during verbal communication, a larger human sample is necessary for an informed conclusion on the utility of these results. It is also necessary to investigate the effect of physical embodiment in a more continuum age group (e.g., kids, adolescents, as well as younger adults between the two age groups included in the present study) and in a more culturally diverse setting.

One crucial difference between our findings and many other previous studies in neuroscience is the choice of summary statistics for the quantification of PFC activation. Whereas previous results are based on averaging-based strategies, the observed bilateral pattern of PFC activation suggests the utility of the variational information of brain activity [61,62,63,64] to address the shortcoming of averaging-based strategies [65,66]. However, further investigation is crucial of the effect of the choice of different summary statistics on the quantitative analyses of the brain activity of humans in basic naturalistic scenarios (e.g., listening to a story narrated by another human) to reevaluate the extent of the underlying emotional and mental reflexes that resulted in such a spread in the frontal brain activity of the individuals in our study. Moreover, such comparative analysis allows a more informed conclusion to be drawn on the observed differences in the frontal activation than with such neuroscientific frameworks as the dedifferentiation hypothesis and its implications on aging brains.

Our results emphasized the potential of physical embodiment for increasing the information content of the frontal brain activity of senior citizens. This observation supports the utility of physically embodied media as a complementary cognitive training strategy for addressing diminished cognitive performance with aging and age-related diseases such as Alzheimer’s and dementia. These observations suggest that the observed effects are potentially inherent to PFC and its responses to social cognition [19]. However, future research with a larger sample of seniors must determine whether the observed effect of the physical embodiment on the information content of their frontal brain activity is sustainable. In addition, we must also verify whether this increase in PFC’s information content improves their cognitive performance.

It is also crucial to compare the effect of physical embodiment with more realistic humanoids as well as other mechanical-looking robots to explore the depths of the potential of the minimalist strategy of the uncanny valley effect. Saygin et al. [67] argued that an android induces a larger suppression effect with a greater predictive coding than humans and humanoid robots. However, their results focused on body movements not conversation or story comprehension scenarios. In this regard, our results indicated the effectiveness of DE for quantification of the PFC responses to minimal anthropomorphic features of a physically embodied medium that induced the feeling of the human presence. An important implication of these results is the potential application of brain information as a promising sensory gateway in the characterization of behavioural responses in human-robot interaction. Therefore, integration of such technologies as NIRS in conjunction with the use of DE in comparative analysis of the effect of different robotic media on PFC activation can determine the utility of DE for evaluation of their design strategy.

In addition, correspondence must be achieved between these responses and communication topics. In particular, future research must determine whether such responses are context-independent or whether their occurrence as well as their degree of variability are affected by the choice of topics. In this respect, our results indicated that DE quantifies the PFC responses of the participants to the story topic in terms of an increase and/or a decrease in information content of this region. Therefore, analysis of the PFC activation in response to the broader topics can determine the ability of DE to quantify their effect on PFC activation which, in turn, allows for modulation of the communicated content through such physically embodied media.

Our study primarily focused on verifying whether the embodiment of a medium bore a significant effect on the brain activation in response to verbally communicated content. Therefore our results, at their present state, are insufficient to allow for drawing informed correspondence between them and the effect of embodiment within the context of social interaction in a broader sense [68]. This limits the utility of our findings for drawing an informed conclusion on different aspects of social interaction (e.g., turn taking (pp. 7–55) [69], gestures and facial expressions [68], effect of the tune and prosody during verbal communication [70], etc.) that might play a relevant and/or more pivotal role in the observed effect of embodiment. For instance, some studies [71] suggested a substantial difference between face-to-face versus mediated contexts due to the effect of the variation in action formation. In the same vein, Rauchbauer et al. [42] compared a human-human interaction (HHI) to human-robot interaction (HRI) and reported that neural markers of mentalizing and social motivation were only present in HHI and not HRI. Therefore, it is important to extend our results to the case of socially interactive and naturalistic scenarios to include fine-grained characterization of verbal and non-verbal behaviours recorded during the interaction to investigate their neural correlates. Such comprehensive analyses are of utmost importance, considering the recent neuroscientific findings that emphasize the fundamental difference between social interaction and social observation [72].

Our results identified the role of physical embodiment in enhancing the PFC activation in response to verbal communication that requires social cognition [19] as well as story comprehension [31]. Moreover, they complemented the findings on its involvement in modulating responses to the anthropomorphic impressions of interacting agents [25,26] by maintaining the same level of conveyed intelligence (i.e., identical stories by the same storyteller). An important factor that demands further scrutiny is the extent of the effect of the varying intelligence in conjunction with the physical embodiment on the dynamics of the observed increase in PFC activation and its information content. As a result, investigation of the effect of these media on the brain activity of human subjects in an interactive setting (e.g., conversations other than listening to a story) is yet another arena for further exploration of the impact of these media and their physical embodiment on our daily lives and well-being.

## 5. Materials and Methods

### 5.1. Participants

Our participants consisted of two groups of individuals: seniors (ten females and ten males, 62–80 years old, MEAN = 70.70, SD = 4.62) and juniors (eleven females and four males, 20–33 years old, MEAN = 22.39, SD = 2.82) groups. Such age separation allowed us to investigate the potential effect of age on the neurological and behavioural responses of the participants to different media. Although the age of the participants is crucial, its significance has been overlooked in highly varying conclusions in previous studies on human-robot interaction [58,59].

All participants were free of neurological or psychiatric disorders and had no history of hearing impairment. Since we failed to collect the NIRS time series of the brain activity of four elderly adults due to recording complications, they were excluded from our analyses. We used an employment service centre for older people and a job-offering site for university students to recruit our participants.

### 5.2. Communication Media

We used an audio speaker, a video-chat system, a humanoid robotic medium and a human (i.e., in-person control condition) to analyze the extent of their effect on the variational information of the participants’ frontal brain activity during the storytelling. Our choice of media reflected the dominant modes of communication in society, including telephone (audio speaker) and video-conferencing (video-chat). We chose a minimalist teleoperated android called Telenoid R4™ (Telenoid hereafter, Figure 7b) [73] to investigate the effect of physical embodiment. Telenoid is approximately 50.0 cm long and weighs about 3.0 kg. It comes with nine degrees-of-freedom (3 for its eyes, 1 for its mouth, 3 for its neck, and 2 for its arms) and is equipped with an audio speaker on its chest. It is primarily designed to investigate the basic and essential elements of embodiment for the efficient representation and transfer of a human-like presence. Therefore, its design follows a minimalist anthropomorphic principle to convey a gender-and-age-neutral look-and-feel. In our present study, we chose a minimalist anthropomorphic embodiment to eliminate the projection of such physical traits as gender and age onto our robotic medium, thereby allowing us to exclusively focus on the utility of minimal anthropomorphic features and their potential effects on the brain activity of human subjects.

Telenoid conveyed the vocal information of its teleoperator through its speaker. Its motion was generated based on the operator’s voice, using an online speech-driven head motion system [74]. However, its eyes and arms were motionless in this study. We placed Telenoid on a stand approximately 1.40 m (Figure 7a) from the participant’s chair to prevent any confounding effect due to tactile interaction (e.g., holding, hugging, etc.). We adjusted this stand to resemble an eye-contact setting between Telenoid and the participant. We maintained the same distance in the case of the other media as well as for the in-person setting. In in-person condition, we adjusted the storyteller’s seat to maintain eye-contact with the participant. For the video-chat, we adjusted its placeholder in such a way that the storyteller’s appearance on the screen resembled an eye-contact setting. In the speaker setting, we placed the video-chat screen in front of the participant (like in the video-chat condition) and placed the speaker behind its screen (Appendix A and Figure A1a–d). We used the same audio device in the speaker and video-chat settings to prevent any confounding effect due to audio quality. We used the same recorded voice of a woman, who was naive to the purpose of this study, in speaker, video-chat, and Telenoid. In the face-to-face setting, the same woman read the stories to the participants.

### 5.3. Sensor Devices

We used near infrared spectroscopy (NIRS) to collect the frontal brain activity of the participants and acquired their NIRS time series data using a wearable optical topography system called “HOT-1000”, developed by Hitachi High-Technologies Corp (Tokyo, Japan). (Figure 8). Participants wore this device on their forehead to record their frontal brain activity through detection of the total blood flow by emitting a wavelength laser light (810 nm) at a 10.0 Hz sampling rate. Data acquisition was carried out through four channels (L1, L3, R1, and R3, Figure 8). Postfix numerical values that are assigned to these channels specify their respective source-detector distances. In other words, L1 and R1 have a 1.0 cm source-detector distance and L3 and R3 have a 3.0 cm source-detector distance. Note that whereas a short-detector distance of 1.0 cm is inadequate for the data acquisition of cortical brain activity using NIRS (e.g., 0.5 cm [75], 1.0 cm [76], 1.5 cm [77], and 2.0 cm [78]), 3.0 cm is suitable [75,77].

To investigate whether the potential changes in the frontal brain activity of the participants were affected by their eye-movement, we also collected their gaze data throughout the storytelling sessions using Tobii Eye Tracker 4C controller. We recorded the eye-movements using pupil centre corneal reflection (PCCR) remote eye-tracking technology at a sampling rate of 90.0 Hz. Prior to gaze data acquisition, we calibrated our device and placed it approximately 80.0 cm from the participant’s seat and 30.0 cm above the ground between the stand of the media and chair of the participants (Figure 7a).

To ensure synchronized data acquisition across sensors, we collected data streams for the video, NIRS, and the gazes through “*labstreaminglayer*” [79] system.

### 5.4. Paradigm

We investigated the effect of three different communication media on the prefrontal cortex activity of the human subjects. Our experimental paradigm consisted of three-minute storytelling sessions in which a woman narrated three-minute stories from Greek mythology through three kinds of communication media: an audio speaker, a video-chat system, and Telenoid. In addition, we considered face-to-face to be a control setting through which we communicated these stories with participants in-person. This resulted in four separate sessions per participant. To prevent any confounding effect of visual distraction and to control the field of view of the participants within the same spatial limit, we placed their seat in a cubicle throughout the experimental sessions (height = 130.0 cm, width = 173.0 cm, depth = 210.0 cm). This cubicle’s side wall is visible in blue in Figure 7a.

Every participant first gave written informed consent in the waiting room next to the experimental room. Then, a male experimenter explained the experiment’s full procedure to the participants. This included the total number of session (four sessions), the duration of the narrated story in each session (i.e., three minutes), instructions about the one-minute rest period prior to the actual session (i.e., sitting still with eyes closed), and the content of the stories. The experimenter also asked the participant to focus on listening to the stories that were either narrated by a woman through a medium or in-person. Next, he led the participants to the experimental room and helped them to get seated in an armchair with proper head support in a sound-attenuated testing chamber. Then the experimenter calibrated the eye tracker device and instructed the participants to fully relax and keep their eyes closed. Last, the experimenter verified the proper adjustment of the medium (or helped the storyteller got comfortable in the proper position for the in-person condition) and began the experimental session. In every session, we first acquired one-minute rest data. Next, the experimenter asked the participants to open their eyes and prepare to listen to the story, followed by a story session during which we recorded the NIRS time series of the frontal brain activity of the participants. During the face-to-face setting, we asked the storyteller to maintain as much eye-contacts with the participants as possible. Figure 7a shows the experimental setup in the Telenoid setting.

At the end of each session, all participants completed questionnaires on their present emotional states. Questions included their perceived level (on an 8-point scale with 1 = not at all and 8 = very much) of the difficulty of the narrated story, their degree of interest in its content, and the induced feeling of fatigue due to listening to it. We also asked about their perceived feeling (on an 8-point scale with 1 = strongly disagree and 8 = strongly agree) of the human presence of the adapted medium in that session. We provided our participants a one-minute rest period prior to the commencement of each of the storytelling sessions and asked them to keep their eyes closed. In this period, we prepared the setting for the next storytelling session. We video-recorded all the activities throughout the experiment.

Every subject participated in all four sessions. We kept the content of the stories intact in these sessions and randomized the order of the media among the participants without changing the order of the narrated stories. The entire experiment lasted about 90 min per participant.

### 5.5. Data Processing

#### 5.5.1. NIRS

First, we baseline-normalized the data by subtracting the mean of the one-minute rest periods. This step that is customary in brain imaging research was to remove the brain activity that was present prior to the start of the task (reflected in the resting period’s recording) and hence did not pertain to the task period’s brain activation. Next and in order to attenuate the effect of systemic physiological artefacts [80] (e.g., cardiac pulsations, respiration, etc.), we applied a one-degree polynomial Butterworth filter with 0.01 and 0.6 Hz for low and high bandpass which was then followed by performing the linear detrending on the data. Detrending of the signal that was adapted from signal processing and time series analysis and forecasting was a necessary step to ensure that the assumptions of stationarity and homoscedasticity (as reflected in wide spread application of linear models in analysis of fNIRS/fMRI time series) were not strongly violated (e.g., due to seasonality and/or repetitive increasing/decreasing patterns). Last, we attenuated the effect of skin blood flow (SBF) using an eigen decomposition technique [81]. This approach considers the first three principal components of the NIRS time series data of the frontal brain activity of the participants during their rest period to represent the SBF. Subsequently, it eliminates the SBF effect by removing these three components from the NIRS time series data of the participants during the task periods. We followed the same approach and removed the first three principal components of the respective rest period of the participants from the NIRS time series of their frontal brain activity that was recorded during the task period (i.e., listening to the stories). Similar to our NIRS recording, [81] also used 3.0 cm source-detector distance channels. Cooper et al. [82] show that this filter also attenuates the motion artefact (e.g., head motion).

A growing body of evidence [61] supports the central role of signal variability in the quantification of the brain’s neurological responses to stimuli and advocates the possibility of the presence of significant information. Miller [62] argued that direct correspondence exists between the amount of information and variance since anything that increases the variance also increases the amount of information. Similarly, Cohen et al. [63] considered the ability to identify meaningful variation in data pertinent to brain activation as an indicator of an effective analysis approach. These results collectively imply that variability in the time series of brain activity efficiently summarizes its responses to a wide range of stimuli. Therefore, given the effect of story encoding and comprehension on frontal brain activity [31,83,84] and considering the involvement of this region in mentalizing physical embodiment [26,27,85], in our experiment we chose to address signal variability to investigate the potential effect of physical embodiment during verbal communication.

The aforementioned findings are in line with the concept of entropy in information theory [86]. These results support the utility of entropy to the solution concept of the decoding of the information content of physiological and neurophysiological signals [87,88]. Extending these empirical results, Keshmiri et al. [64] proved that differential entropy (DE, i.e., entropy of a continuous random variable) quantifies the information content of the frontal brain activity in terms of a shift (e.g., increase and/or decrease) in its variational information. Given these observations, we adapted DE to summarize the variational information of the brain activity of the participants in our study.

We computed the DE, $H(X_{i})$, of a three-minute time series of the frontal brain activity of the *i*th participant as [86]:(1)$$H\left(X_{i}\right)=\frac{1}{2}\log_{2}\left(2\pi e\sigma_{X_{i}}^{2}\right),\;i=1,\ldots ,N,$$where $\sigma_{X_{i}}^{2}$ is the variance of the NIRS time series of the frontal brain activity of the *i*th participant and $H(X_{i})$ gives the average variational information of the frontal brain activity of the *i*th participant.

#### 5.5.2. Gaze Data

We first acquired the portion of these data per medium and per participant that fell within the pre-determined elliptical area that surrounded the spatial location of the storyteller’s face. We chose this elliptical area based on the storyteller’s chair in a face-to-face location or her image’s location on the screen for the video-chat and the speaker (Appendix A, Figure A1b–d). We placed this elliptical area around Telenoid’s face (Figure 7a). Given its location, we considered a 14.0 and 16.0 cm elliptical area as its short and long diameters around its face location. Next, we formulated a gazing metric (eye aversion ratio (EAR)) to determine the relative deviation of the visual attention of the participants from the media during the storytelling session:(2)$$EAR=1.0-\frac{C}{T}$$where C is the number of gazing data points that fell within the pre-determined elliptical area (described above) per medium and per participant. T which is a hypothetical maximum number of gaze data points of the participants in the absence of any eye-aversion is calculated as T=eyetrackersamplingrate×storytellingtime=90.0Hz×180.0s=16,200.00maximumgazedatapoints. EAR resembles an explicit eye-movement measurement that quantifies (0 ≤ EAR ≤ 1) the deviation of the visual attention of the participants from media or the human speaker. EAR→0 as C→T, implying the least aversive behaviour and/or deviation of the eye-field-of-view from the media or the human speaker. It is also worth noting that EAR restricts the legitimate gaze of the participants to data points that fall within the face area in media/in-person, suggesting that the remainder of the gaze data of the participants indicates eye-movement/aversion. Therefore, a strongly conservative metric ensures the eye-movements of the participants without underestimating the potential effect on PFC activation.

### 5.6. Statistical Analyses

We analyzed the effect of the ages of the participants by the variational information of their frontal brain activity with a two-way mixed-design ANOVA with age (i.e., elderly and younger participants) and media factors (i.e., speaker (S), video-chat (V), Telenoid (T), and face-to-face (F)). We used the Mendoza multisample sphericity test for this purpose. When sphericity was violated, we adjusted the degrees of freedom with Greenhouse-Geisser’s epsilon and conducted post hoc comparisons with Shaffer’s modified sequentially rejective Bonferroni correction.

For the analysis of the self-assessed responses of the participants to their perceived feelings of the human presence, we used a two-way mixed-design ANOVA (age and media factors) with the Mendoza multisample sphericity test. When sphericity was violated, we adjusted the degrees of freedom with Greenhouse-Geisser’s epsilon and conducted post hoc comparisons with Shaffer’s modified sequentially rejective Bonferroni correction. We used Gramm [89] for data visualization (SM 2, SM 3, and SM 4 for these results with respect to other self-assessed responses). Prior to the two-way mixed-design ANOVA analyses, we checked our data for normality using a one-sample Kolmogorov-Smirnov test. All media passed this test, except for DE and the self-assessed feelings of the human presence by elderly adults in face-to-face setting.

We analyzed the potential correspondence between the self-assessed responses of the participants to their feelings of the human presence and the induced variational information in their frontal brain activity (i.e., DE) by adapted media using Pearson correlations (for the Pearson correlation results with respect to other self-assessed responses as well as analyses of the self-assessed responses, see Appendix B, Appendix C, Appendix D, Appendix E, Appendix F, Appendix G).

For the gaze data, we utilized EAR, as formulated in Equation (Equation 2), to determine any potential confounding effect of the eye-movements of the participants on the information content of their frontal brain activity (i.e., DE). We calculated the Pearson correlations between the EAR of the participants and the information content of their frontal brain activity. Additionally, we computed the Pearson correlation between EAR and the self-assessed responses of the participants to their feelings of the human presence (SM 7 for the Pearson correlation results with respect to other self-assessed responses) to determine whether their gazing patterns revealed any potential preference toward embodiment and/or anthropomorphic impressions (e.g., view of the storyteller in video-chat or her actual presence face-to-face). The existence of a potentially significant correlation between EAR and the self-assessed responses of the participants might indicate the sufficiency and reliability of their behavioural responses for the evaluation of the effect of physical embodiment, discarding the necessity of the analysis of frontal brain activity as a unique neurophysiological marker of such an effect.

We used Python 2.7 for data acquisition and processing. We carried out a shift function, a difference asymmetry function, and Pearson correlation analyses in Matlab R2016a and used R 3.3.2 for parametric and non-parametric mixed-design analyses.

## Figures and Tables

**Figure 1 entropy-21-00875-f001:**
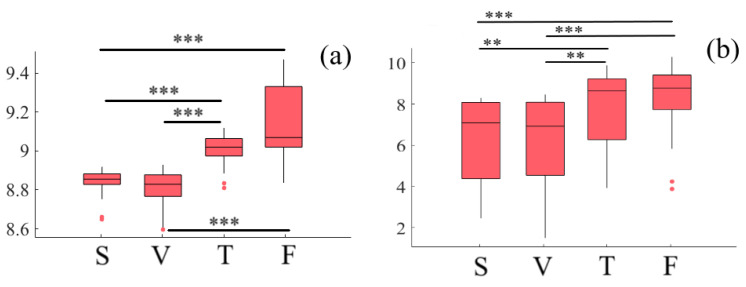
Descriptive statistics of post hoc comparison on DE of (**a**) left-hemispheric and (**b**) right-hemispheric frontal brain activity of participants in different media settings. Asterisks mark media with significant differences (*: *p* < 0.05, **: *p* < 0.01, ***: *p* < 0.001).

**Figure 2 entropy-21-00875-f002:**
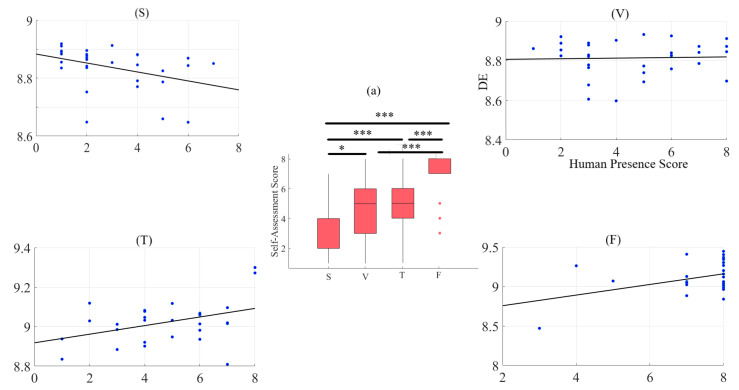
(**a**) Descriptive statistics of post hoc comparison with Shaffer’s modified sequentially rejective Bonferroni correction of self-assessment of participants to induced sense of human presence in different media settings. Asterisks mark media with significant differences (*: *p* < 0.05, **: *p* < 0.01, ***: *p* < 0.001). Subplots (S), (V), (T), and (F): Pearson correlation between variational information of left-hemispheric frontal brain activity of human subjects and their self-assessed responses to induced feelings of human presence by speaker (S), video-chat (V), Telenoid (T), and face-to-face (F).

**Figure 3 entropy-21-00875-f003:**
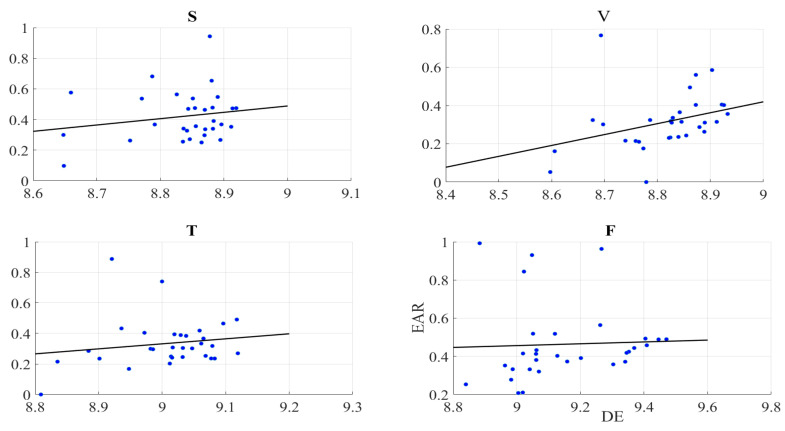
Pearson correlation between left-hemispheric frontal brain activity (i.e., DE) of participants and eye aversion ratio (EAR) of their gazing data: speaker (S), video-chat (V), Telenoid (T), and face-to-face (F).

**Figure 4 entropy-21-00875-f004:**
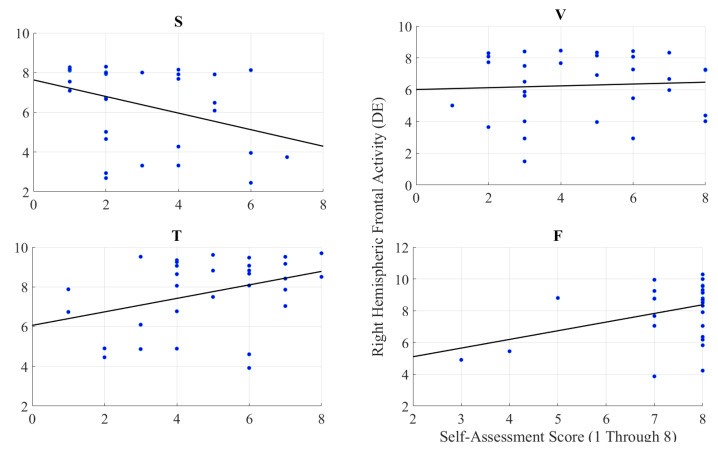
Pearson correlation between variational information of right-hemispheric frontal brain activity of human subjects and their self-assessed responses to induced feelings of human presence by speaker (S), video-chat (V), Telenoid (T), and face-to-face (F).

**Figure 5 entropy-21-00875-f005:**
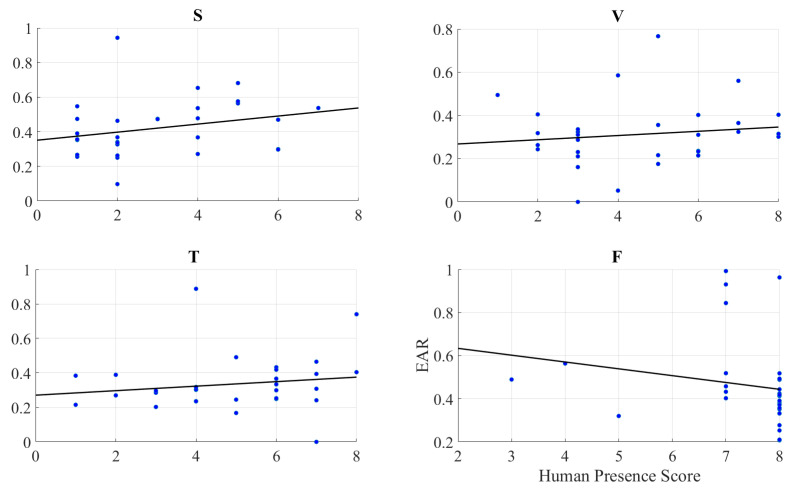
Pearson correlation between self-assessed responses of participants to feelings of presence and eye aversion ratio (EAR) of their gazing data: speaker (S), video-chat (V), Telenoid (T), and face-to-face (F).

**Figure 6 entropy-21-00875-f006:**
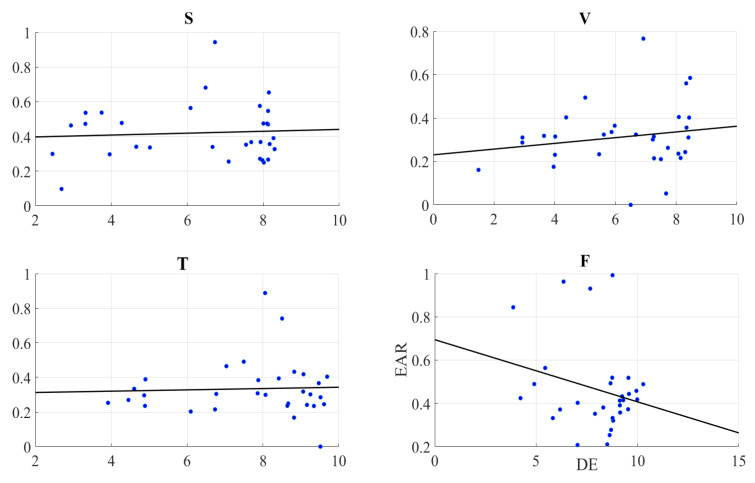
Pearson correlation between information content (i.e., DE) of the right-hemispheric frontal brain activity of participants and their eye aversion ratio (EAR) in speaker (S), video-chat (V), Telenoid (T), and face-to-face (F).

**Figure 7 entropy-21-00875-f007:**
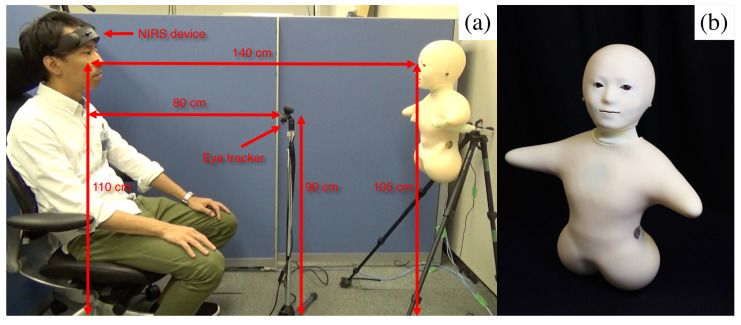
(**a**) Experimenter demonstrates experimental setup. (**b**) Telenoid. Figures for other settings (i.e., speaker, video-chat, and face-to-face) are provided in Appendix A.

**Figure 8 entropy-21-00875-f008:**
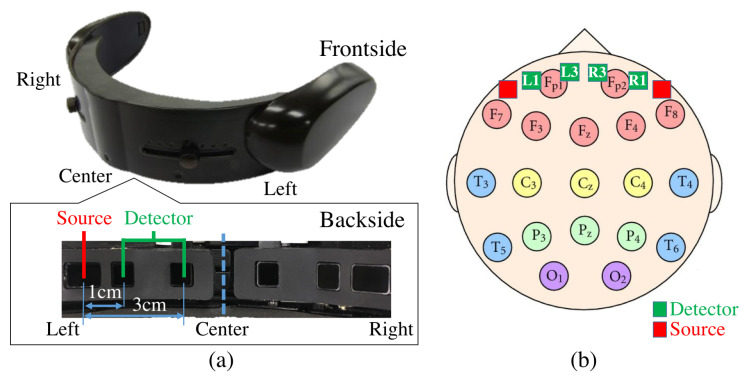
(**a**) Near infrared spectroscopy (NIRS) device in present study. Bottom subplot on left shows arrangement of source-detector of four channels of this device. Distances between short (i.e., 1.0 cm) and long (i.e., 3.0 cm) source and detector of left and right channels are shown. (**b**) Arrangement of 10–20 International Standard System: In this figure, relative locations of channels of NIRS device in our study (i.e., L1, L3, R1, and R3) are depicted in red (i.e., sources) and green (i.e., detectors) squares. L1, R1, L3, and R3 are channels with short (i.e., 1.0 cm) and long (i.e., 3.0 cm) source-detector distances.

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
