# Peer review of "Differential Effect of the Physical Embodiment on the Prefrontal Cortex Activity as Quantified by Its Entropy"

_entropy, 2019, doi:10.3390/e21090875_

Round 1

Reviewer 1 Report

Paper is difficult for a researcher who has worked for naturalistic studies of social interaction. Paper is very weak in naturalistic details, strong in quantified measurements.

The aspects of interaction trough which meaning of action is formed in activities incl. story telling are

1) turn taking

Sacks, H., Schegloff, E. A., & Jefferson, G. (1978). A simplest systematics for the organization of turn taking for conversation. In Studies in the organization of conversational interaction (pp. 7-55). Academic Press.

2) gaze. e.g.,

Rossano, F. (2013). 15 Gaze in Conversation. The handbook of conversation analysis, 308.

3) Facial expressions, gestures

Streeck, J., Goodwin, C., & LeBaron, C. (Eds.). (2011). Embodied interaction: Language and body in the material world. Cambridge University Press.

4) Prosody

Selting, M. (1996). On the interplay of syntax and prosody in the constitution of turn-constructional units and turns in conversation. Pragmatics. Quarterly Publication of the International Pragmatics Association (IPrA), 6(3), 371-388.

To analyze, which aspects of social interaction are relevant, it were useful to analyze how story telling participates in action formation:

Mandelbaum, J. (2013). 24 Storytelling in Conversation. The handbook of conversation analysis, 492.

there are also studies on how social interaction is modified in face-to-face and in mediated contexts., suggesting that the relevant difference is variation in action formation:

Arminen, I., Licoppe, C., & Spagnolli, A. (2016). Respecifying mediated interaction. Research on Language and Social Interaction, 49(4), 290-309.

Also the difference between reading and story telling were relevant:

Person, R. F. (2015). From Conversation to Oral Tradition: A Simplest Systematics for Oral Traditions. Routledge.

If and only if relevant factors of social interaction were taken into consideration, then more detailed and relevant study design could be implemented so that salient features of interaction could be analysed and their variation in different contexts could be taken into consideration.

Paper is problematic from the point of view of studies of social interaction and exisiting systematic knowledge in the field that is completely neglected.

Author Response

We agree with the reviewer's comprehensive comment on the importance of the social interaction and the effect of such paradigm on formation of the individuals’ perception about both content of the communication and the medium (whether another person or a medium). We also realize that storytelling, as a form of communication, is indeed a type of interaction. Moreover, and as adequately noted by the reviewer, the degree of naturalness (i.e., naturalistic settings) plays a pivotal role in formation of such perceptions. Although some of the aspect of such physical interaction (e.g., gaze, feeling of fatigue, feeling of human presence, etc.) are reported in the present manuscript (human presence and gaze Sections2.2.2 and 2.2.3, gaze analysis in Section 2.3.,andself-assessed responses of the participants Appendices A through G.), it is important to note that the present study was not designed for analysis of such interactions. In particular, in the present study, we primarily attempted to verify whether the effect of embodiment on the verbally communicated content and irrespective of the level of interaction and naturalness of such an interaction. This was the reason that we prevented (as much as it was possible that is) any type of physical interaction between the participants and the adapted media in this study (for details, see Section 5.2. in the current version of the manuscript). To further clarify this point, we added the following to the Introduction (lines 104-108):

It is also crucial to note that the present study was not to analyze the effect of embodiment on the human-medium interaction [4244] and instead focused on verifying whether the embodiment of a medium bore a significant effect on the brain activation in response to verbally communicated content. To this end, we prevented any physical contact between the participants and the media in this study (see Section 5.2, for details).”

In this regard, our two recent results (i.e., citations [43] and [44]) appear to extend the findings in the current manuscript to the case of human-robot conversation. However, further analyses are necessary prior to drawing a definite conclusions.

Next and in order to emphasize the limitation of our study from the social interaction perspective and to hint at the future direction of this research, we added the following paragraph to the Section 4. Concluding Remarks (lines 372-387).

Our study primarily focused on verifying whether the embodiment of a medium bore a significant effect on the brain activation in response to verbally communicated content and therefore, at their present state, insufficient to allow for drawing informed correspondence between them and the effect of embodiment within the context of social interaction in a broader sense [68]. This limits the utility of our findings for drawing an informed conclusion on different aspects of social interaction (e.g., turn  taking [p. 7-55][69], gestures and facial expressions [68], effect of the tune and prosody during verbal communication [70], etc.) that might play a relevant and/or more pivotal role in the observed effect of embodiment. For instance, some studies [71] suggested substantial difference between face-to-face versus mediated contexts due to the effect of the variation in action formation. In the same vein, Rauchbauer et al. [42] compared a human-human interaction (HHI) to human-robot interaction (HRI) and reported that neural markers of mentalizing and social motivation were only present in HHI and not HRI. Therefore, it is important to extend our results to the case of socially interactive and naturalistic scenarios to include fine-grained characterization of verbal and non-verbal behaviours recorded during the interaction to investigate their neural correlates. Such comprehensive analyses are of utmost important, considering the recent neuroscientific findings that emphasize the fundamental difference between social interaction and social observation [72].

Reviewer 2 Report

In this paper the Authors after a review of Computer-mediated-communication (CMC) research, are proposing to study the importance of the physical embodiment of conversational partners.

The results provide evidence that communicating through a physically embodied medium affects the humans’ frontal brain activity whose patterns potentially resemble those of in-person communication.

I found that this paper is very interesting and that the obtained results are very promising, however in order to further improve I would suggest to change the structure of the paper following these scheme:

Introduction Materials and methods Results Discussion Concluding Remarks

Please check that some figures are in the references.

I also recommend to improve the conclusions and more references on the background

Author Response

Reviewer Comment:I would suggest to change the structure of the paper following these scheme: Introduction Materials and methods.

Authors’ Response:We have used the LateX template that is provided by MDPI Entropy Journal and followed the formatting layout in this template while preparing our manuscript. To further ensure that we did not use an outdated template, we also checked the other published articles by Journal Entropy which appear to follow the same formatting structure. However, we can change the manuscript’s structure and reformat its write-up in case the reviewer finds this change necessary.

Reviewer Comment: Results Discussion Concluding Remarks Please check that some figures are in the references.

Authors’ Response: We have rearranged the figures and ensured that figures are no longer extended to the “Discussion” and “Reference” Sections.

Reviewer Comment:I also recommend to improve the conclusions and more references on the background

Authors’ Response: We extended the current version of the manuscript as follows:

Line104-108: We clarified that our study was designed to primarily examine the potential effect of the medium embodiment on verbally communicated content and not the level of interaction between the human subjects and these media. Subsequently, we cited threerecent studiesthat focused on the correspondence between media-embodiment and interaction.

Lines121-141: We further extended the current views on the importance of the study of the media embodiment, citing adequate results. We closed this paragraph by highlighting how our results contribute to such studies.

Lines 372-387:We explained the limitation of our results with regard to social aspect of human-robot interaction, citing relevant results in the literature in this regard and hinting at some of the aspect of such analyses in the future.
